# Cannabinoid Mixture Affects the Fate and Functions of B Cells through the Modulation of the Caspase and MAP Kinase Pathways

**DOI:** 10.3390/cells12040588

**Published:** 2023-02-11

**Authors:** Marie-Claude Lampron, Isabelle Paré, Mohammed Al-Zharani, Abdelhabib Semlali, Lionel Loubaki

**Affiliations:** 1Héma-Québec, Medical Affairs and Innovation, 1070 Avenue des Sciences-de-la-Vie, Québec, QC G1V 5C3, Canada; 2Department of Biology, College of Science, Imam Mohammad Ibn Saud Islamic University (IMSIU), Riyadh 11623, Saudi Arabia; 3Groupe de Recherche en Écologie Buccale, Faculté de Médecine Dentaire, Université Laval, Québec, QC G1V 0A6, Canada; 4Department of Biochemistry, Microbiology and Bioinformatics, Laval University, Québec, QC G1V 0A6, Canada

**Keywords:** B cells, cannabinoids, apoptosis, caspase, oxidative stress

## Abstract

Cannabis use is continuously increasing in Canada, raising concerns about its potential impact on immunity. The current study assessed the impact of a cannabinoid mixture (CM) on B cells and the mechanisms by which the CM exerts its potential anti-inflammatory properties. Peripheral blood mononuclear cells (PBMCs) were treated with different concentrations of the CM to evaluate cytotoxicity. In addition, flow cytometry was used to evaluate oxidative stress, antioxidant levels, mitochondrial membrane potential, apoptosis, caspase activation, and the activation of key signaling pathways (ERK1/2, NF-κB, STAT5, and p38). The number of IgM- and IgG-expressing cells was assessed using FluoroSpot, and the cytokine production profile of the B cells was explored using a cytokine array. Our results reveal that the CM induced B-cell cytotoxicity in a dose-dependent manner, which was mediated by apoptosis. The levels of ROS and those of the activated caspases, mitochondrial membrane potential, and DNA damage increased following exposure to the CM (3 µg/mL). In addition, the activation of MAP Kinase, STATs, and the NF-κB pathway and the number of IgM- and IgG-expressing cells were reduced following exposure to the CM. Furthermore, the exposure to the CM significantly altered the cytokine profile of the B cells. Our results suggest that cannabinoids have a detrimental effect on B cells, inducing caspase-mediated apoptosis.

## 1. Introduction

Cannabis is the most widely used recreational drug in Canada. According to the Canadian Alcohol and Drugs Survey (CADS), the prevalence of cannabis use increased from 11% to 19% between 2017 and 2019 [1]. Cannabis is a complex plant containing ∼500 phytochemicals, of which at least 60 belong to the phytocannabinoid class. These phytochemical compounds have shown therapeutic benefits as analgesics, anti-inflammatory agents, anti-emetics, and anticonvulsive agents, and they can improve muscle tone, mood state, cognition, and appetite [2,3].

Cannabinoids act through the endocannabinoid system (ECS), which includes the cannabinoid type 1 (CB1) and type 2 (CB2) receptors, their endogenous ligands (endocannabinoids), and the enzymes responsible for their synthesis and degradation [4]. Cannabinoids also modulate several non-cannabinoid receptors and ion channels, and they act through various receptor-independent pathways—for example, by delaying the reuptake of endocannabinoids and neurotransmitters (such as anandamide and adenosine) and by enhancing or inhibiting the binding of certain G-protein-coupled receptors to their ligands [4]. 

Cannabinoids affect various physiological processes, including the immune response. However, the effect of cannabinoids on immune cells is not well understood due to conflicting evidence. Cannabinoids have been shown to hinder the migration of leukocytes and the production of reactive oxygen species (ROS), and they have been shown to induce oxidative stress and the release of pro-inflammatory cytokines [5,6,7,8,9]. They can also limit the ability of macrophages to produce nitric oxide and IL-6 in response to lipopolysaccharides (LPSs), induce apoptosis in B and T cells, and reduce the cytolytic activity of natural killer (NK) cells [10,11,12,13,14]. However, cannabinoids have also been shown to stimulate the inflammatory response (mainly through their metabolites, which increase the secretion of some pro-inflammatory cytokines [15,16,17,18]) and promote the biosynthesis of eicosanoids, such as prostaglandins and leukotrienes (which are important mediators of inflammation) [19,20]. Of note, conflicting results have been reported regarding the effects of cannabinoids on B cells. El-Gohary et al. [21] reported that the oral ingestion of cannabis decreases the number of B cells, the serum levels of immunoglobulins (IgG and IgM), and the levels of the C3 and C4 complement proteins. By contrast, other studies have found no change in the number of B cells, an increase or decrease in IgE levels [22,23], a decrease in serum IgG levels, and an increase in IgD levels, with various impacts on IgA and IgM secretion [24,25]. Furthermore, one study reported that THC can cause a dose-dependent increase in B-cell proliferation [10,26], whereas other studies found that cannabinoids hinder B-cell proliferation in response to LPSs [26,27,28,29]. As we recently reported that a cannabinoid mixture (CM) can impair the quality of red blood cells (RBCs) and platelets by triggering RBC hemolysis and reducing platelet aggregation [30], we therefore wanted to assess the impact of exposure to a CM on B cells, as they produce antibodies, are involved in antigen presentation, and strongly express CB2 receptors (CB2Rs) [28], making them a cornerstone of the immune response.

## 2. Materials and Methods

### 2.1. Isolation and Storage of Peripheral Blood Mononuclear Cells

This study was approved by Héma-Québec’s Research Ethics Committee (CER#2020-010), and all participants signed an informed consent form. Whole blood (450 mL) was collected using the Leukotrap^®^ WB system (Haemonetics, Braintree, MA, USA) according to the manufacturer’s instructions. Immediately after the blood donation, PBMCs were isolated using gradient centrifugation with a Ficoll-Paque solution (Cytiva, Vancouver, BC, Canada) and Leucosep tubes (Greiner Bio-One; Monroe, NC, USA) according to the manufacturer’s instructions. PBMCs were collected and washed with DPBS (Thermo Fisher Scientific, Waltham, MA, USA) supplemented with 0.25% human albumin (CLS Behring, Ottawa, ON, Canada). The PBMCs were then suspended in a Plasma-Lyte solution (Baxter, Mississauga, ON, Canada) containing 5% human albumin and 18% CryoSure-Dex40 (WAK-Chemi medical, Steinbach, Germany), aliquoted, and frozen in liquid nitrogen for subsequent use.

### 2.2. Cell Culture and PBMC Exposure to a Cannabinoid Mixture

The PBMCs were thawed using the ThawSTAR™ system (BioLife Solutions, Bothell, WA, USA) and suspended in RPMI 1640 media (Thermo Fisher Scientific) supplemented with 20% fetal bovine serum (FBS; Thermo Fisher Scientific) and penicillin–streptomycin 1X (PEN/STREP; Sigma-Aldrich, St-Louis, MO, USA), and they were centrifuged for 10 min at 600× *g*. The supernatant was discarded, and the cells were suspended in RPMI/20% FBS + PEN/STREP 1X at 1 × 10^6^ cells/mL. The cells were then seeded in a 12-well plate (Sigma-Aldrich) and incubated for 3 hours at 37 °C/5% CO_2_ to enable adherent cells to adhere. Then, different concentrations (ranging from 1 to 24 µg/mL) of the CM-8 components (#C-219, Cerilliant, Round Rock, TX, USA) or equivalent volume of methanol (MeOH; vehicle in which the CM is dissolved) were added to the required experimental conditions and incubated overnight at 37 °C/5% CO_2_. This incubation time was based on the former blood donation deferral time after cannabis consumption that was used in our institution.

### 2.3. Cytotoxicity Assay

Following the cell culture and exposure to the CM, PBMC cytotoxicity was assessed through the quantification of lactate dehydrogenase (LDH) levels in the cell culture supernatant. Briefly, the PBMCs were suspended in RPMI/1% FBS + PEN/STREP 1X at 1 × 10^5^ cells/mL and seeded in a 12-well plate (Sigma-Aldrich). The cells were then incubated for 3 hours at 37 °C/5% CO_2_, before adding the CM or equivalent volume of MeOH, followed by an overnight incubation at 37 °C/5% CO_2_. The cell supernatants were then collected for LDH quantification using a CyQUANT™ LDH Cytotoxicity Assay Kit (Cat#C20300, Fisher Scientific) according to the manufacturer instructions. 

### 2.4. Apoptosis and CD45+/CD19+ Cell Count

In addition to cytotoxicity, PBMC apoptosis was assessed by using flow cytometry employing an Allophycocyanin (APC) conjugated-Annexin V Apoptosis Detection Kit (Cat# 640932; BioLegend, San Diego, CA, USA) according to the manufacturer’s instructions. Furthermore, the number of CD45+/CD19+ cells was measured in each experimental condition of the dose-response assay. Briefly, following the overnight exposure to the CM, 50 × 10^3^ cells were collected, washed in PBS, and stained with a fluorescein isothiocyanate (FITC)-conjugated anti-CD19 antibody (Clone HIB19; BD Biosciences, Franklin Lakes, NJ, USA) and an APC-conjugated anti-CD45 antibody (Clone HI20; BD Biosciences). Data were acquired using a BD Accuri™ C6 flow cytometer (BD Biosciences) and analyzed using FCS Express™ 6 software (De NovoSoftware, Los Angeles, CA, USA). Our gating strategy was as follows: first, all CD45+ cells were identified in a dot plot, and then CD19+ cells were identified in this population and the number of events enumerated. The acquisition volume was 200 µL.

### 2.5. Assessment of CB2R Expression

Along with the apoptosis assessment, the expression of CB2R was evaluated. Briefly, following the overnight exposure to the CM, 50 × 10^3^ cells were collected, washed in PBS, and stained with a fluorescein isothiocyanate (FITC)-conjugated anti-CD19 antibody (Clone HIB19; BD Biosciences, Franklin Lakes, NJ, USA) and an Alexa Fluor^®^ 647-conjugated Human Cannabinoid R2/CB2/CNR2 Antibody (Clone 352110R; R&D system). Data were acquired using the BD Accuri™ C6 flow cytometer (BD Biosciences) and analyzed using FCS Express™ 6 software (De NovoSoftware, Los Angeles, CA, USA). Our gating strategy was as follows: first, all CD19+ cells were identified in a dot plot, and then CB2R+ cells were identified in this population and the number of events enumerated. 

### 2.6. Oxidative and Anti-Oxidative Stress Responses

To assess changes in the oxidative and anti-oxidative stress responses in B cells, PBMCs (with or without exposure to 3 µg/mL CM) were collected and stained with an APC-conjugated, anti-CD19 antibody (BD Biosciences) and exposed to CellROX™ Oxidative Stress Reagents (Cat# C10492, Thermo Fisher Scientific) according to the manufacturer’s instructions. In addition, the anti-oxidative response was assessed by measuring the intracellular levels of glutathione (GSH) using an intracellular glutathione assay (Cat#9137, ImmunoChemistry, Davis, CA, USA) along with APC-conjugated, anti-CD19 antibody staining (BD Biosciences), according to the manufacturer’s instructions. All data were acquired with the BD Accuri™ C6 flow cytometer (BD Biosciences) and analyzed using FCS Express™ 6 software (De Novo software). Our gating strategy was as follows: all cell populations were gated, and CD19+/ROS+ cells were identified in this population.

### 2.7. Mitochondrial Membrane Potential 

To assess alterations in the mitochondrial membrane potential of B cells, PBMCs (with or without exposure to 3 µg/mL CM) were collected and stained using a MitoProbe™ DiOC2(3) Assay Kit (Cat# M34150, Thermo Fisher Scientific) along with an APC-conjugated, anti-CD19 antibody (BD Biosciences). All data were acquired with the BD Accuri™ C6 flow cytometer (BD Biosciences) and analyzed using FCS Express™ 6 software (De Novo software).

### 2.8. Apoptosis PCR Array

To investigate the differential expressions of apoptosis-related genes after exposure to the CM, real-time PCR was performed on purified B cells using an RT2 Profiler PCR Array for Human Apoptosis (cat#330231; PAHS-012ZD-6; Qiagen; Mississauga, ON, Canada). Briefly, PBMC (suspended in RPMI/20% FBS + PEN/STREP 1X at 1 × 10^6^ cells/mL) were seeded in a 12-well plate (Sigma-Aldrich), incubated for 3 h at 37 °C/5% CO_2_, and subsequently exposed to 3 µg/mL of the CM (Cerilliant) followed by an overnight incubation at 37 °C/5% CO_2_. After the incubation, CD19+ cells were isolated using an EasySep™ Human B Cell Isolation Kit (cat#17954; Stemcell Technologies, Vancouver, BC, Canada) according to the manufacturer’s instructions. Following isolation, the B cells from each experimental condition were transferred into a lysis buffer for total RNA extraction using a RiboPure™- Blood kit; (cat#AM1928; Thermo Fisher Scientific). The concentration and purity of the total RNA were assessed using a NanoDrop spectrophotometer (Thermo Fisher). The RNA was reverse-transcribed into cDNAs using a cDNA conversion kit (RT2 First Strand Kit; cat#330401; Qiagen), and a PCR array was performed using the SYBR Green master mix (cat#330504; Qiagen) along with an RT2 profiler plate. This plate was centrifuged for 1 min at 1000× *g*/25 °C, and real-time PCR was carried out with a CFX-96 Real-Time PCR Detection System (Bio-Rad; Mississauga, ON, Canada) according to the manufacturer’s instructions. The array measures the expressions of 84 key genes involved in apoptosis. Cycle threshold (CT) values were analyzed using the data analysis web portal at http://www.qiagen.com/geneglobe (accessed on 31 January 2023). The samples were assigned to control (untreated) and test groups (methanol and CM). The CT values were normalized based on a manual selection of reference genes. The data analysis web portal calculates the fold change/regulation using the delta-delta CT method, in which delta CT is calculated between a gene of interest and the average of several reference genes, followed by delta-delta CT calculations (∆ CT (Test Group)-∆ CT (Control Group)). The fold changes are then calculated as follows: 2^(−∆∆CT). The data analysis web portal also plots the heat map.

### 2.9. Assessment of Caspase Activation 

To detect the activated caspases in the B cells, a Calbiochem^®^ Caspase Detection Kit (FITC-VAD-FMK, cat#QIA90, Sigma-Aldrich) was used along with an APC-conjugated, anti-CD19 antibody (BD Biosciences). Briefly, PBMCs were cultured with or without 3 µg/mL CM as described above. Following the culture, the PBMCs were stained with the components of the detection kit and an anti-CD19 antibody (BD Biosciences). In addition, Z-VAD-FMK—a cell-permeable, irreversible pan-caspase inhibitor provided in the kit—was used to inhibit the caspase processing and apoptosis induced by the CM in the B cells. Data were acquired with the BD Accuri™ C6 flow cytometer (BD Biosciences). The gating strategy was as follows: first, all CD19+ cells were selected in a dot plot, and the activated caspases were identified in this CD19+ population.

### 2.10. Assessment of DNA Damage 

Following the overnight exposure of the PBMCs to the CM (as described above), the PBMCs were collected and diluted at 5 × 10^5^ cells/mL. Then, 50 µL of the cell suspension (25 × 10^3^ cells) was transferred into flow cytometry tubes, DPBS + 2% FBS was added, and the cells were centrifuged for 5 min at 500× *g* at room temperature (RT). The cells were then fixed with a 1.5% paraformaldehyde solution (Sigma-Aldrich) for 20 min at RT, followed by a second wash with DPBS + 2% FBS. After the fixation step, the cells were permeabilized by exposing them to 90% methanol for 20 min at 4 °C. After another wash with DPBS + 2% FBS, the cells were stained with 5 µL of an Alexa Fluor^®^ 488-conjugated, γH2AX antibody (Clone N1-431; BD Biosciences) and 20 µL of an APC-conjugated, CD19 antibody (BD Biosciences) for 20 min at RT. A final wash with DPBS + 2% FBS was performed before acquiring the data with the BD Accuri™ C6 flow cytometer (BD Biosciences). The gating strategy was as follows: first, all CD19+ cells were selected in a dot plot, and γH2AX+ cells were then identified in this CD19+ population.

### 2.11. Analysis of Cell Signaling Pathways 

The signaling pathways involved in the development and functions of B cells were assessed using flow cytometry with the same protocol used to assess the DNA damage (described above). The only difference was the antibody used along with an FITC-conjugated, anti-CD19 antibody (BD Biosciences). For the assessment, 25 × 10^3^ cells were stained using the following Alexa Fluor^®^ 647 conjugated antibodies from BD Biosciences: phospho-NF-κB (Ser529; clone B33B4WP; Thermo Fisher Scientific), phospho-ERK1/2 (clone 20A), phospho-p38 (clone pT180/py182), and phospho-STAT5 (clone pY694). Data were acquired with the BD Accuri™ C6 flow cytometer. The gating strategy was as follows: cells were identified as CD19+ cells, and the cells positive for the aforementioned targets were identified in this CD19+ population. The data were analyzed using FCS Express™ 6 software (De Novo Software).

### 2.12. Detection and Enumeration of IgG-/IgM-Secreting B Cells

To evaluate the effect of the CM on B cells, a Human IgG/IgM Dual-Color B cell FluoroSpot kit (ELDB8079NL, R&D systems, Minneapolis, MN, USA) was used to quantify the IgG-/IgM-secreting cells according to the manufacturer’s instructions. In total, 10^4^ viable PBMCs (exposed or not exposed to the CM) were plated in each well of a 96-microplate provided in the kit. Images (6 pictures/well per condition) were acquired using an Olympus BX53 microscope equipped with a DP80 digital camera (Olympus, Shinjuku-ku, Tokyo, Japan) and they were analyzed using cellSens imaging software (Olympus).

### 2.13. Cytokine Expression Profiles of B Cells 

Following exposure to the CM, the PBMCs were collected, and the B cells were isolated using an EasySep™ Human B Cell Isolation Kit (Stemcell Technologies) as described above. The B cells were then lysed in a cell lysis mix (Thermo Fisher Scientific), and the lysate was frozen for cytokine profiling. The amount of protein in each cell lysate was determined using a BCA Protein Assay (Thermo Fisher). The resulting data were used to normalize the results of the cytokine array, which was performed by Eve technologies Corp (Calgary, AB, Canada) using a Human Cytokine/Chemokine 71-Plex Discovery Assay^®^ (HD71). 

### 2.14. Statistical Analysis

All analyses were performed using GraphPad Prism 9.2.0 (GraphPad, San Diego, CA, USA). All values are reported as means ± standard errors of the mean, and statistical comparisons were carried out using a paired *t*-test, a Kruskal–Wallis test (i.e., a nonparametric ANOVA), a Wilcoxon matched-pairs signed-ranks test, or a Mann–Whitney test (where applicable). A *p*-value below 0.05 was considered statistically significant.

## 3. Results

### 3.1. Exposure to the Cannabinoid Mixture Favors B-Cell Apoptosis

Cannabinoids are detectable in the plasma within a few seconds after the first inhalation, and their bioavailability following inhalation varies widely (i.e., 2–56%), in part due to intra- and inter-subject variabilities in smoking dynamics causing some uncertainty in dose delivery [31,32]. Thus, doses were selected considering bioavailabilities of 2% (1 µg/mL), 6% (3 µg/mL), 12.5% (6 µg/mL), 25% (12 µg/mL), and 50% (24 µg/mL)—the equivalent of smoking 1 g of cannabis containing 24% of cannabinoids. This dose response revealed a dose-dependent increase in LDH levels, suggesting a significant cytotoxic effect of the CM on the PBMCs (Figure 1A). An Annexin V/PI analysis confirmed a dose-dependent increase in cell death in two specific PBMC subpopulations (P2 and P3) (Figure 1B,C), including B cells. The number of B cells was also significantly reduced at CM concentrations of at least 3 µg/mL (Figure 1D), which was associated with a significant decrease in the expression of CB2R (Figure 1E). 

### 3.2. Exposure to the Cannabinoid Mixture Significantly Reduces the Mitochondrial Membrane Potential of B Cells 

We recently reported that a CM significantly increases the levels of ROS in oral cancer cells [33]. Therefore, we looked at ROS as a potential mechanism involved in the CM-induced death of B cells. From here on, we used the concentration of 3 µg/mL because, at this dose, we had a significant number of both living cells and cells undergoing apoptosis and not only dead cells. Thus, the cells were stained with an anti-CD19 antibody and a ROS marker. Exposure to 3 µg/mL CM significantly increased the intracellular levels of ROS in the CD19+ cells (Figure 2A), but it did not affect the anti-oxidative response (as measured via GSH levels) (Figure 2B). 

Increased levels of ROS have been associated with mitochondrial damage [34]. Thus, using a MitoProbe™ DiOC_2_(3) Assay Kit, we measured the mitochondrial membrane potential of the B cells (CD19+ cells). The CM also reduced the median fluorescence intensity (MFI) of the B cells by 10-fold (untreated: 797,704.9 ± 228,317.31; methanol: 839,641.2 ± 237,704.82 MFI; CM-treated: 79,188.6 ± 25,246.65 MFI) (Figure 2C). 

### 3.3. The Cannabinoid Mixture Induces B-Cell Apoptosis through the Caspase Pathway 

To identify the pathways involved in the CM-induced apoptosis of the B cells, we analyzed the expressions of 84 genes involved in apoptosis using a PCR array. The PBMCs were exposed to 3 µg/mL of the CM overnight, the B cells were isolated using an EasySep™ Human B Cell Isolation Kit, and their RNA was extracted. cDNAs were generated, and a real-time PCR was performed according to the manufacturer’s instructions. Among the 84 genes analyzed, 3 were upregulated by ≥3-fold after exposure to the vehicle: LTBR (3.55-fold), TNFRSF1A (3.15-fold), and TNFRSF25 (4.34-fold). By contrast, after exposure to the CM, 27 were significantly upregulated compared with the untreated B cells: BAG1 (3.63-fold), BCL2L10 (7.07-fold), BIK (7.07-fold), BIRC5 (7.07-fold), CD40LG (5.63-fold), CIDEA (7.07-fold), CRADD (7.07-fold), CYCS (3.17-fold), DAPK1 (7.07-fold), FADD (3.20-fold), FASLG (7.07-fold), GADD45A (4.38-fold), HRK (3.60-fold), IL10 (3.10-fold), LTA (4.48-fold), LTBR (3.85-fold), NOL3 (7.07-fold), TNF (5.85-fold), TNFRSF11B (7.07-fold), TNFRSF21 (13.11-fold), TNFRSF25 (8.95-fold), TNFSF8 (3.74-fold), TP73 (7.07-fold), TRADD (7.07-fold), TRAF3 (3.00-fold), and both CASP14 (7.07-fold) and CASP5 (7.07-fold) (Figure 3A). 

Based on this finding, we explored the caspase pathway given its role in the initiation and execution of cell death [35]. A flow cytometry analysis of the activated caspases revealed a 4-fold increase in the level of activated caspase in the CM-treated B cells (22,068.7 ± 4433.30 MFI) compared to the untreated (5211.3 ± 1119.55 MFI) or vehicle-treated B cells (6424.3 ± 1900.61 MFI) (Figure 3B). In addition, we found that the CM potently increased the expression of the DNA damage marker γH2AX in the B cells by ~4-fold (untreated: 6.9 ± 0.95% of positive cells; methanol: 8.4 ± 1.42; CM: 28.7 ± 5.43% of positive cells; Figure 3C).

To confirm that caspase activation is involved in the CM-induced apoptosis of the B cells, PBMCs were treated with the pan caspase inhibitor Z-VAD-FMK and exposed to the CM. CM-induced B-cell death was significantly abrogated by the caspase inhibitor (untreated: 63.0 ± 10% of living cells; methanol: 63.2 ± 10.48% of living cells; CM: 19.4 ± 5.65; Z-VAD-FMK+ CM: 44.6 ± 11.63% of living cells, Figure 4A–C).

### 3.4. The Cannabinoid Mixture Affects B-Cell Signaling Pathways and Function

ERK, NF-κB, STAT5, and p38 have been shown to play important roles in B-cell proliferation, differentiation, survival, and immunoglobulin production [36,37,38,39]. Thus, we explored the phosphorylation of these key signaling molecules. The CM reduced the phosphorylation of ERK1/2 (untreated: 61% ± 6.96% of positive cells; methanol: 58.45% ± 5.75; CM: 18.7% ± 9.03% of positive cells); of NF-κB (untreated: 68.8% ± 3.60% of positive cells; methanol: 67.6% ± 5.31; CM: 18.1% ± 3.22% of positive cells); of STAT5 (untreated: 63.1% ± 5.34% of positive cells; methanol: 53.4% ± 2.95; CM: 7.5% ± 1.8% of positive cells); and of p38 (untreated: 32.4% ± 4.75% of positive cells; methanol: 37.7% ± 5.83%; CM: 8% ± 2.65% of positive cells; Figure 5A), which was associated with a significant reduction in the number of IgM- and IgG-expressing B cells (Figure 5B–D).

### 3.5. The Cannabinoid Mixture Affects the Cytokine Secretion Profile of B Cells

Using a Human Cytokine/Chemokine 71-Plex Discovery Assay^®^ (HD71), we analyzed how the CM affects the cytokine secretion profile of B cells. Our results reveal that twenty-one (21) cytokines, including IL-6 and TNF-α, were not detected. Eighteen (17) cytokines were not modulated (*n* = 17), while thirteen cytokines (13), including RANTES and IL-27, were downregulated. Finally, eight (8) cytokines, including IFN-α2 and CXCL9, were upregulated (*n* = 5) (Figure 6).

## 4. Discussion

We recently reported that a CM impairs the quality of RBCs and platelets by triggering RBC hemolysis and reducing platelet aggregation [30]. However, to the best of our knowledge, the effects of a CM on B-cell cytotoxicity and the pathways driving cell death have not been investigated. Here, we showed that exposing PBMCs to various concentrations of a CM (1–24 µg/mL) caused significant cytotoxicity to B cells and limited their ability to produce immunoglobulins. 

Cannabinoids are detectable in the plasma within a few seconds after the first inhalation, and their concentration peaks within 3-10 minutes [40,41]. They are quickly eliminated by pyrolysis, and their bioavailability following inhalation varies widely (i.e., 2−56%) [31,32,42]. For example, it has been found that, six minutes after the inhalation of 13 mg of THC (i.e., 2.5 × 10^19^ THC molecules), only 2.8% of this THC (1.4 × 10^14^ molecules/mL) is detectable in the plasma [31]. Moreover, it has also been found that, one minute after intravenously administering a single bolus of 5 mg of THC (i.e., 9.55 × 10^18^ molecules), the plasma concentration of THC is 4.28 × 10^14^ molecules/mL [43], suggesting the rapid elimination of cannabinoids and, more importantly, a large difference between the administered dose and that measured in the blood. 

Given this evidence, we selected concentrations corresponding to bioavailabilities of 2% (1 µg/mL), 6% (3 µg/mL), 12.5% (6 µg/mL), 24% (12 µg/mL), and 50% (24 µg/mL), which equate to inhaling a joint containing 1 g of cannabis with 24% of cannabinoids. Using this range of concentrations, the CM induced cytotoxicity in a dose-dependent manner in the PBMCs, particularly in the B cells. Indeed, we observed that the CM significantly reduced the number of B cells, which is consistent with a previous study conducted by El-Gohary et al. [21], who reported that cannabis users have lower PBMC, T-cell, B-cell, and NK-cell counts than non-users. This cytotoxicity may be explained by the expressions of the cannabinoid receptors on immune cells, particularly that of CB2R, which is known to mediate most of the immunosuppressive effects of the ECS [20,44] and is highly expressed on the surface of B cells [28]. This hypothesis is further supported by the significant reduction in the expression of CB2R following the exposure to the CM, suggesting that its activation occurs through an internalization mechanism, as reported by Atwood et al. [45]. 

Based on the dose response of the CM, we decided to continue all our experiments with the concentration of 3 µg/mL because this allowed us to have both living cells and cells undergoing apoptosis, which is in line with our goal of documenting the mechanism involved in CM-induced B-cell death. Thus, to explore the mechanisms involved in CM-induced B-cell death, we assessed the production of ROS, since cannabinoids have been shown to induce apoptosis in human monocytes through an increased production of ROS and the disruption of mitochondrial integrity [46]. Thus, as expected, the CM significantly reduced the mitochondrial membrane potential, which resulted in a mitochondrial free radical leak and, thus, explains the increase in the levels of intracellular ROS. Of interest, a similar increase in oxidative stress has been reported in synthetic cannabinoid users [47]. In addition to the increase in the production of ROS, no modulation of the anti-oxidative response (GSH) was observed, which suggests that an imbalance in redox homeostasis is a mechanism involved in the B-cell apoptosis observed herein. Indeed, the expressions of 27 genes involved in apoptosis were significantly altered after exposure to the CM, including those of caspases, which we further explored given their roles in the initiation and execution of cell death [35,48]. As expected, the CM significantly increased the levels of activated caspases. Further, apoptosis was abrogated when the PBMCs were co-treated with the pan caspase inhibitor Z-VAD-FMK, thus confirming the roles of caspases in the CM-induced apoptosis of the B cells. However, this inhibition was not complete, which suggests the involvement of other mechanisms in addition to caspases. This hypothesis is supported by the findings obtained by McKallip et al. [14], who reported that the exposure of mice (in vivo) to a pan caspase inhibitor prior to THC administration partially blocked the apoptotic effects of THC, which also suggests the involvement of mechanisms other than caspases, as well as supporting the findings of our apoptosis gene array.

The study of the intracellular signaling pathways in the B cells, carried out to characterize the molecular events responsible for the functional modifications that are elicited in these cells following exposure to the CM, revealed a reduction in the levels of phosphorylated ERK1/2, NF-κB, STAT5, and p38, which are involved in B-cell proliferation, differentiation, survival, and immunoglobulin production [36,37,38,39]. More specifically, ERK1/2 phosphorylation is one of the signaling events that canonically occurs following CB2R stimulation by an agonist; it may be considered a biomarker to verify CB2R activation, and it also plays a key role in the efficient generation of IgG-bearing B cells by promoting their survival [36,49]. In addition, the phosphorylation of NF-κB, which is important for B-cell maturation and activation, mediated through the B-cell receptor (BCR) [39,50], was reduced by exposure to the CM, thus suggesting the potential ability of cannabinoids to impair B-cell activation. Furthermore, the phosphorylation of STAT5 and p38, which have been reported to regulate B-cell proliferation and survival, was also reduced following exposure to the CM [37,38]. Together, these data could explain the significantly reduced number of IgG- and IgM-expressing cells reported herein, which was associated with the downregulated levels of cytokines that are involved in B-cell immunoglobulin production, such as RANTES and IL-27 [51,52], while some cytokines, such as IFN-α2, which lower the threshold for B-cell activation, were upregulated, probably as a compensatory mechanism [53]. We can also talk about another potential compensatory mechanism which is the increase in the expression of CXCL9 with the aim to maintain the expression of its receptor CXCR3 given its role in the differentiation of B cells [54].

Taken together, these results suggest that cannabinoids exert negative impacts on key aspects of B-cell fate and the immune response in general. They also raise concerns about the ability of cannabis users to effectively fight certain infections. Indeed, cannabis users have been reported to have a higher risk of fungal exposure (as cannabis may contain inhalable Aspergillus organisms) and infection associated with an increased variety of immunologic lung disorders [55,56,57]. Furthermore, these detrimental effects were observed following exposure to a concentration as low as 3 µg/mL, which suggests, as also observed by Melèn et al. [58], that even occasional cannabis users might exhibit (transitory) B-cell death.

## 5. Conclusions

To the best of our knowledge, this study provides the first evidence that a brief (in vitro) exposure of PBMCs to a CM impairs their survival and function. Specifically, the CM triggered B-cell death in a dose-dependent manner, despite being rapidly eliminated. Although our in vitro model likely reproduces several features of cannabis use, further studies conducted in cannabis users are required to confirm and better understand the impact of cannabis use on the immune system.

## Figures and Tables

**Figure 1 cells-12-00588-f001:**
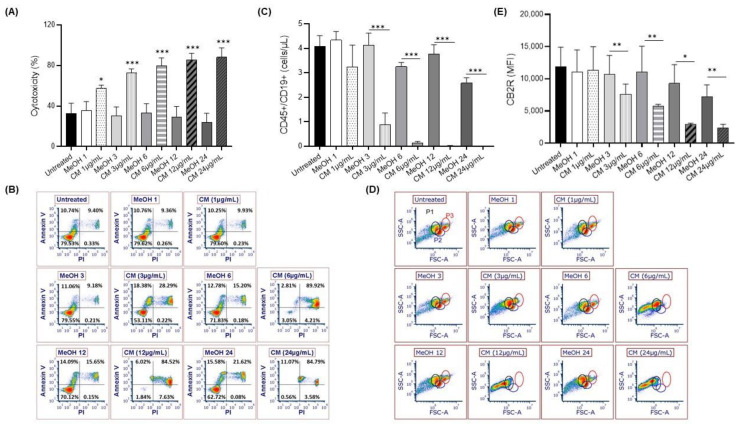
Cytotoxicity and Annexin V/PI expression after exposing PBMCs to a cannabinoid mixture. (**A**) PBMCs were exposed (or not) to different CM concentrations (1–24 µg/mL), and cytotoxicity was assessed using a lactate dehydrogenase assay. (**B**) Representative flow cytometry results of Annexin V/PI expression on PBMCs, which was measured to assess apoptosis following exposure to different doses of CM. (**C**) Representative flow cytometry results of the FSC/SSC profile of PBMCs following exposure to different CM concentrations. (**D**) Enumeration of B cells (CD45+/CD19+ cells) following exposure to different CM concentrations. (**E**) Expression of CB2R on B cells (CD19+/CB2R+ cells) following exposure to different CM concentrations. Data are presented as means and standard errors of the mean. * *p* ˂ 0.05; ** *p* ˂ 0.001; *** *p* ˂ 0.0001. *n* = 5 experiments. CM = cannabinoid mixture; FSC = forward scatter; MeOH = methanol; SSC = side scatter. P1, P2, and P3 represent different populations of PBMCs.

**Figure 2 cells-12-00588-f002:**
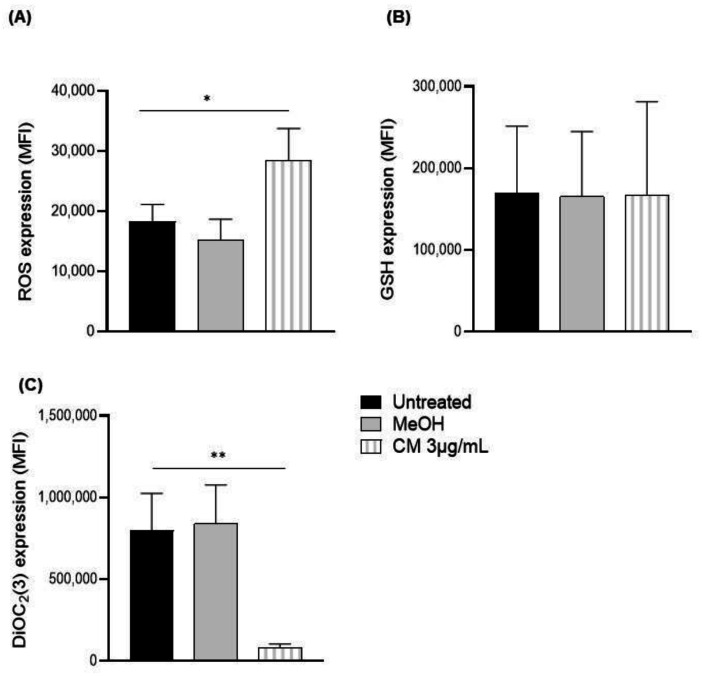
Response to oxidative and anti-oxidative stress, as well as mitochondrial membrane potential after exposing PBMCs to a cannabinoid mixture. PBMCs were exposed (or not) to 3 µg/mL of CM and stained with an APC-conjugated, anti-CD19 antibody and (**A**) treated with CellROX™ Oxidative Stress Reagents to measure ROS levels or (**B**) exposed to intracellular glutathione assay components to measure the anti-oxidative response via flow cytometry. (**C**) The mitochondrial membrane potential of CD19+ cells was also assessed via flow cytometry using a MitoProbe™ DiOC_2_(3) Assay Kit. Data are presented as means and standard errors of the mean. * *p* ˂ 0.05; ** *p* ˂ 0.001. *n* = 5 experiments. CM = cannabinoid mixture; GSH = glutathione; MeOH = methanol; MFI = median fluorescence unit; ROS = reactive oxygen species.

**Figure 3 cells-12-00588-f003:**
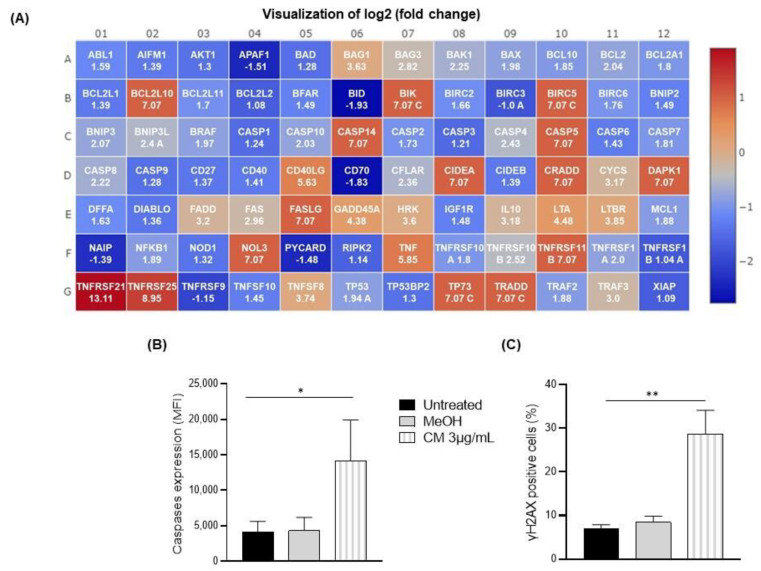
Apoptosis and DNA damage after exposing PBMCs to a cannabinoid mixture. PBMCs were exposed (or not) to 3 µg/mL of the CM, and B cells were isolated using an EasySep™ Human B Cell Isolation Kit. (**A**) Heatmap of CM compared to the untreated condition. (**B**) Activated caspase levels in CD19+ cells were assessed via flow cytometry using a Caspase Detection Kit. (**C**) DNA damage, as measured by the expression of γH2AX, was assessed via flow cytometry. Data are presented as means and standard errors of the mean. *: *p* ˂ 0.05, **: *p* ˂ 0.001. *n* = 5 experiments. CM = cannabinoid mixture; MeOH = methanol; MFI = median fluorescence unit.

**Figure 4 cells-12-00588-f004:**
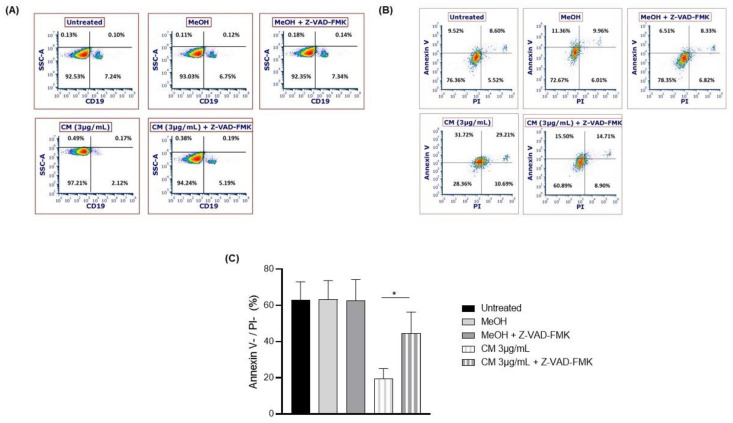
Death of B cells after co-treatment with the caspase inhibitor Z-VAD-FMK and the cannabinoid mixture. PBMCs were exposed (or not) to the caspase inhibitor Z-VAD-FMK and subsequently exposed (or not) to 3 µg/mL of the CM. After exposure to the CM, PBMCs were stained with an anti-CD19 antibody and (**A**) the persistence of the B-cell population, and (**B**,**C**) the levels of non-apoptotic cells (i.e., Annexin V/PI levels) were measured. Data are presented as means and standard errors of the mean. *: *p* ˂ 0.05. *n* = 5 experiments. CM = cannabinoid mixture; FSC = forward scatter; MeOH = methanol; SSC = side scatter.

**Figure 5 cells-12-00588-f005:**
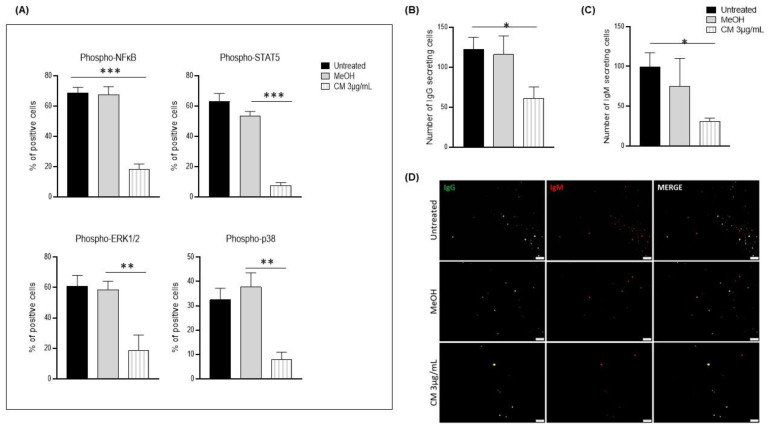
MAP Kinase, NF-κB, and STAT signaling pathways after exposing PBMCs to the cannabinoid mixture. PBMCs were exposed (or not) to 3 µg/mL of the CM and co-stained with an APC-conjugated, anti-CD19 antibody coupled with Alexa Fluor^®^ 647-conjugated phospho-NF-κB, phospho-ERK1/2, phospho-p38, or phospho-STAT5 antibodies. (**A**) Representative histograms of the expressions of the aforementioned proteins with and without CM exposure. Number of (**B**) IgG-secreting cells and (**C**) number of IgM-secreting cells were quantified using FluoroSpot. (**D**) Representative fluorescence microscopy image of IgG- and IgM-secreting cells. Data are presented as means and standard errors of the mean. *: *p* ˂ 0.05. ** *p* ˂ 0.001; *** *p* ˂ 0.0001. *n* = 5 experiments. CM = cannabinoid mixture; MeOH = methanol; UT = untreated.

**Figure 6 cells-12-00588-f006:**
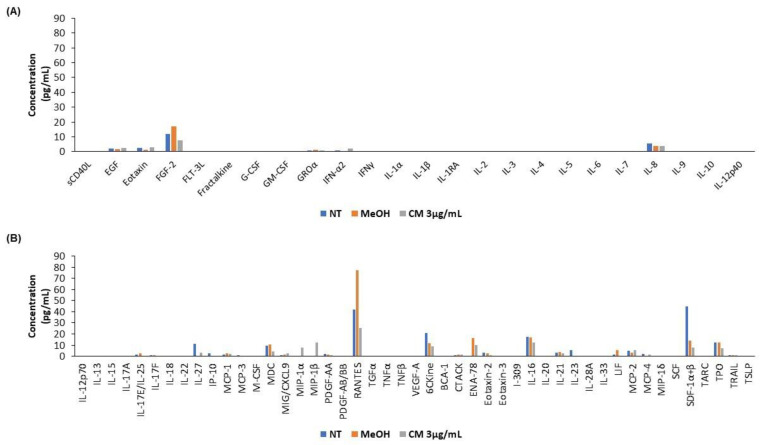
Cytokine profile after exposing B cells to the cannabinoid mixture. PBMCs were exposed (or not) to 3 µg/mL of CM, and B cells were isolated using an EasySep™ Human B Cell Isolation Kit. B cells were then lysed, and a cytokine array was performed. (**A**,**B**) Representative histograms of the expression of the 71 proteins tested by the cytokine array.

## Data Availability

Not applicable.

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
