# Peer review of "Cannabinoid Mixture Affects the Fate and Functions of B Cells through the Modulation of the Caspase and MAP Kinase Pathways"

_cells, 2023, doi:10.3390/cells12040588_

Round 1

Reviewer 1 Report

Article: cells-2144878

Cannabinoid mixture affects the fate and functions of B cells through the modulation of

the caspase and MAP Kinase pathways.

In the presented study, the authors describe the effect of a cannabinoid mixture (CM) on B cells and the mechanisms by which a CM exerts its potential anti-inflammatory properties. Investigation of the anti-inflammatory activity of cannabinoid ligands is an interesting subject. However, revision of the current manuscript is needed to clarify some points.

1. The authors could expand the reported literature by citing other papers describing the activity of synthetic or newly synthesized cannabinoids on lymphocytes or PBMCs, such as: doi.org/10.3390/molecules27010064; doi: 10.1124/jpet.102.033506; doi: 10.1016/j.clim.2006.11.002.

2. The authors show the results of treatment only following to an overnight exposure, there are data or some information about longer CM incubation times? Usually in other works, results of cytotoxic analysis and/or pro-apoptotic effect are shown up to 72 hours of treatment. (10.1002/ijc.30483; doi:10.3390/ijms19071958).

3. Although all the analyzes were performed using a cytofluorimetric technique are still well performed and interesting, in any case the cell interaction of this mixture of cannabinoids is not very specified, some demonstration can be added with respect to the involvement of cannabinoid receptors (CBRs), probably in the cells used we can think of the CB2R?? In several papers the CBR involvement was demonstrated by using CBR inhibitor.

4. The quality of all the Figures must be implemented.

5. The References could be updated with some more recent papers.

Author Response

  1. The authors could expand the reported literature by citing other papers describing the activity of synthetic or newly synthesized cannabinoids on lymphocytes or PBMCs, such as: doi.org/10.3390/molecules27010064; doi: 10.1124/jpet.102.033506; doi: 10.1016/j.clim.2006.11.002.

We thank the reviewer for this comment.  Recommended papers have been added in the manuscript: for Mc Kallip et al. (reference #14), Capozzi et al. (reference #44) and Lomabard et al (reference #48). In addition, more recent references have been added both in the introduction (page 2) and discussion section (Page11 and 13).

  1. The authors show the results of treatment only following to an overnight exposure, there are data or some information about longer CM incubation times? Usually in other works, results of cytotoxic analysis and/or pro-apoptotic effect are shown up to 72 hours of treatment. (10.1002/ijc.30483; doi:10.3390/ijms19071958).

This study is a follow-up to one we recently published on the effect of CM on red blood cells and platelets (Lampron et al. Blood Transfus. 2022 Aug 1. doi: 10.2450/2022.0100-22.). Indeed, the choice to incubate all night is based on the deferral time for blood donation that existed in our blood bank (until 12h post consumption) and which allowed us to show that even such a brief exposure had important consequences on the functionality of platelets and red blood cells. This is also where the originality of our work lies because we show that despite exposure as brief as all night, knowing that the cannabinoids are eliminated quickly (in less than an hour), significant effects can still be observed. In addition, our study evaluated different levels of bioavailability of the product which came into contact with the cells.  This is more clearly stated in the manuscript in section 2.2.Cell culture and PBMC exposure to a cannabinoid mixture on page 2 line 96-98.

  1. Although all the analyzes were performed using a cytofluorimetric technique are still well performed and interesting, in any case the cell interaction of this mixture of cannabinoids is not very specified, some demonstration can be added with respect to the involvement of cannabinoid receptors (CBRs), probably in the cells used we can think of the CB2R?? In several papers the CBR involvement was demonstrated by using CBR inhibitor.

We agree with the reviewer. We have added information regarding the level of CB2R on CD19+ following exposition to a CM. Data have been added on page 5 line 251. figure 1 have been modified to include data of the expression of CB2R (page 6). The importance of these data are discussed in the manuscript on page 11, line 499-504.   

  1. The quality of all the Figures must be implemented.

The reviewer is right. We have improved the quality of the figures by converting them to high resolution (600 DPI).

  1. The References could be updated with some more recent papers.

As recommended by the reviewer, we have added more recent references including those proposed by the reviewer on page 2 (introduction) and on page 11-13 (Discussion).

Reviewer 2 Report

Referee’s comment

Article n° Cells - 2144878

Title: Cannabinoid mixture affects the fate and functions of B cells through the modulation of the caspase and MAP Kinase pathways

Authors: Lampron MC,  Paré I, Al- Zharani M, Semlali A, Loubaki L.

General comment

In this manuscript Lampron et al. performed a study in which they investigated the effect of cannabinoid mixture (CM) treatment in peripheral blood mononuclear cells (PBMC). They provided to evaluate different parameters such as apoptosis, ROS production, DNA damage, mitochondrial membrane, cytokine production etc. By the obtained results, authors concluded that CM treatment significantly down-regulated the cytokine production of PBMC. Therefore they suggest that cannabinoids produce a deleterious effect on these cells, also inducing caspase-mediated apoptosis. 

The paper is of interest and in general was well organized and experiments conducted in the right way. Therefore I believe that a revision is necessary and below there are some  suggestions to make the paper more suitable.

1)     The authors don't appear to be very familiar with the pharmacology. Initially they rightly used doses of CM equivalent to those inhaled by smokers. Then, however, they continued the experiments using only the 3 mg/ml dose of CM. In this way it is not possible to appreciate a possible dose-dependent effect which underlies the functioning of a preparation on a given parameter. This point represents a gap in this paper.

2)     It would be very interesting to understand which of the 8 compounds in the mixture had the greatest effect in the variations of the parameters used.

3)     In the text it is reported that the key signalling pathways ERK1/2, NF-κB, STAT5 and p38 were inactivated by BM treatment. Actually, this is not apparent from the data reported. Although some cytokines linked to these pathways have been analysed, this proves nothing, as other pathways may have been triggered. Involvement of these pathways involves direct assessment of ERK-1/2, p38 MAPK and NF-kB directly without intermediates.

4)     It is strange that the authors did not take into account the most important cytokines that mediate the inflammatory process, such as IL-1beta, TNF-alpha and IL-6 instead of other, albeit important, less involved ones.

5)     Cannabinoids, as reported in the text, act on specific receptors which normally increase their concentration after interaction with the ligand. The evaluation of the receptor mRNA concentrations would have demonstrated the direct action of CM on the specific receptors.

6)     The authors continue to ignore the most basic rules of pharmacology. In fact, the equivalent use of Met-OH in terms of quantity is not understood. Are the numbers reported ml, or ml? Since Met-OH is the solvent of CM, it is sufficient to use it only once, for each parameter, as a control in volume equal to that in which administered CM was dissolved.

7)     The Discussion section is too short for how many results have been achieved. All the results must be better discussed and better compared with the previous ones, justifying the different data in a solid way. For example, some cytokines were found to be up-regulated, the justification reported is that could be a compensatory mechanism. It refers only to IFN-α2, and the other 4 cytokines?

Author Response

1)     The authors don't appear to be very familiar with the pharmacology. Initially they rightly used doses of CM equivalent to those inhaled by smokers. Then, however, they continued the experiments using only the 3 mg/ml dose of CM. In this way it is not possible to appreciate a possible dose-dependent effect which underlies the functioning of a preparation on a given parameter. This point represents a gap in this paper.

      We thank the reviewer for these comments. As mentioned in the article, we tested different bioavailabilities and those above 3µg/mL proved to be very harmful for the cells. Our interest was to understand the mechanisms that lead to cell death and this objective could not have been achieved by analyzing only dead cells. This is the reason why we decided to continue our work with the concentration of 3µg/mL (about IC50) because it allowed us to have both living cells and cells in the process of apoptosis rather than dead cells. This was clarified in the manuscript section 3.2. on page 7 line 290-292.

 2)     It would be very interesting to understand which of the 8 compounds in the mixture had the greatest effect in the variations of the parameters used.

      As a first step, we wanted to test a research product that was as close as possible to cannabis (instead of one specific cannabinoid) and which contains more than one cannabinoid. Thus, we agree with the reviewer, it would be really interesting and we intend to do this investigation very soon in order to determine which component(s) is or are responsible for the effect that is observed in the current manuscript. In addition, Dr Semlali recently demonstrated that cannabinoids (Δ9-THC and Δ8-THC) were able to decrease oral cancer cell growth through various mechanisms, including apoptosis, autophagy, and oxidative stress (Abdelhabib Semlali  et al 2021. DOI: 10.1016/j.archoralbio.2021.105200) and that these two compounds presented a high synergistic effect. This could be a good starting point.

3)     In the text it is reported that the key signalling pathways ERK1/2, NF-κB, STAT5 and p38 were inactivated by BM treatment. Actually, this is not apparent from the data reported. Although some cytokines linked to these pathways have been analysed, this proves nothing, as other pathways may have been triggered. Involvement of these pathways involves direct assessment of ERK-1/2, p38 MAPK and NF-kB directly without intermediates.

The reviewer is right, our images does not reflect very well the differences that we have highlighted. Indeed, we have shown for ERK1/2 ( Untreated: 61% ± 6.96% of positive cells; Methanol: 58.45% ± 5.75; CM: 18.7% ± 9.03% of positive cells; i.e. approximately 3 times less activation); of NF-κB (Untreated: 68.8% ± 3.60 % of positive cells; Methanol: 67.6% ± 5.31; CM: 18.1% ± 3.22 % of positive cells, i.e. approximately 4 times less activation); of STAT5 (Untreated: 63.1% ± 5.34 % of positive cells; Methanol: 53.4% ± 2.95; CM: 7.5% ± 1.8% of positive cells, i.e. approximately 8 times less activation) and of p38 (Untreated: 32.4% ± 4.75% of positive cells; Methanol: 37.7% ± 5.83%; CM: 8% ± 2.65% of positive cells; i.e. 4 times mins of activation). These data are mentioned in the article section 3.4. on page 9 and stated in a clearer way in histograms (Figure 5 on page 10).

4)     It is strange that the authors did not take into account the most important cytokines that mediate the inflammatory process, such as IL-1beta, TNF-alpha and IL-6 instead of other, albeit important, less involved ones.

We understand the questioning of the reviewer. However, the cytokines mentioned were not detectable or weakly detected in our cytokines array. To respond to the reviewer concern, the entire cytokine array is now presented as heat map in Figure 6 on page 11.

5)     Cannabinoids, as reported in the text, act on specific receptors which normally increase their concentration after interaction with the ligand. The evaluation of the receptor mRNA concentrations would have demonstrated the direct action of CM on the specific receptors.

We agree with the reviewer, unfortunately we did not evaluate the expression of the mRNA of the

CBRs but we evaluated the expression of CB2R by flow cytometry given it’s important role in the immunosuppressive effect of cannabinoids (Lombard et al.Clinical Immunology 2007, 122, 259–270, doi:10.1016/j.clim.2006.11.002). CB2R results have been added on figure 1, panel E on Page 6) and this result is also discussed in the manuscript on page 11(line 514 to 519).

6)     The authors continue to ignore the most basic rules of pharmacology. In fact, the equivalent use of Met-OH in terms of quantity is not understood. Are the numbers reported ml, or ml? Since Met-OH is the solvent of CM, it is sufficient to use it only once, for each parameter, as a control in volume equal to that in which administered CM was dissolved.

Information provided by the manufacturer indicate that the final concentration of methanol in which CM is dissolved is 90%. Thus, we dispensed the equivalent volume (in mL) of methanol at 90% as a control. we thought, that was the right way to do it. In addition, the dose response showed that all concentrations of methanol used in our experiments did not affect the cells supporting the fact that methanol at the concentration we used it was not detrimental to the cells. For example, for 24 µg/mL of a CM (at 500µg/mL), which was equivalent to 12µL of a CM in 2 mL of culture medium, we added the same of methanol 90% meaning 12 µL as a control. We hope this clarify the reviewer concern.

7)     The Discussion section is too short for how many results have been achieved. All the results must be better discussed and better compared with the previous ones, justifying the different data in a solid way. For example, some cytokines were found to be up-regulated, the justification reported is that could be a compensatory mechanism. It refers only to IFN-α2, and the other 4 cytokines?

As requested by the reviewer, we have extended our discussion. See section 4. Discussion on page 11. In addition, as shown in figure 6 (cytokine array) a lot of cytokine were not detectable or had a very low level of expression. So, we chose these cytokines (IL-27, RANTES, INF-α2 and CXCL9) based on their level of expression which we consider as significant (this choice is totally arbitrary; other people might have chosen to focus on other cytokines). The level of expression of the other cytokines being very low, this leads us to make no assumptions about the impact of their modulation in the biology of B cells. So, we chose the aforementioned cytokines because of their role in B cell biology which is the focus of our article.

Round 2

Reviewer 1 Report

Thanks for your revised version of the paper.

All my requests have been satisfied, thus the manuscript may be accepted.

Reviewer 2 Report

The manuscript appears to have been sufficiently improved.